# Utility of a multiplex pathogen detection system directly from respiratory specimens for treatment and diagnostic stewardship

Smriti Srivastava,[1] Neha Sharad,[1] Vandana Vijayeta Kiro,[1] Aparna Ningombam,[2] Sharad Shrivastava,[2] Kamran Farooque,[3] Purva Mathur[2]

**ABSTRACT** The availability of syndrome-based panels for various ailments has widened the scope of diagnostics in many clinical settings. These panels can detect a multitude of pathogens responsible for a particular condition, which can lead to a timely diagnosis and better treatment outcomes. In contrast to traditional identification methods based on pathogen growth on culture, syndrome-based panels offer a quicker diagnosis, which can be especially beneficial in situations requiring urgent care, such as intensive care units. One such panel is the Biofire Filmarray Pneumonia plus Panel (BFP), which we have compared against microbiological culture and identification. The lower respiratory samples from patients were tested with BFP, culture, and identification with culture considered the gold standard. The phenotypic antibiotic susceptibility results (Vitek 2) were compared with the antimicrobial resistance (AMR) genes detected in BFP. Statistical analysis was carried out using GraphPad 7.0 and MS Excel (Microsoft Inc.). The results showed a positive percent agreement of 100% and a negative percent agreement of 47.8% with an overall agreement of 76.72% compared to culture. BFP was better at identifying fastidious bacteria, and the agreement with culture was higher for high bacterial identification numbers ($10^7$ and $10^6$). There was also a correlation between the number of pathogens detected and growth in culture. Carbapenemase genes were detected in around 80% of phenotypically resistant samples and correlated with in-house PCR 60% of the time. Hence, BFP results need to be interpreted with caution especially when multiple pathogens are detected. Similarly, the presence or absence of AMR genes should be used to guide the therapy while being watchful of unusual resistance or susceptibility. The cost constraints and low throughput call for patient selection criteria and prioritization in emergency or resource-limited conditions.

**IMPORTANCE** Application of syndrome-based panels in clinical microbiology is of huge support in infectious conditions requiring urgent interventions, such as pneumonia. Interpreting the results requires caution; hence, we have compared the results obtained from Biofire Filmarray Pneumonia plus Panel with standard microbiological methods.

**KEYWORDS** pneumonia, Biofire, culture, PCR, AMR, endotracheal aspirate, bronchoalveolar lavage

Lower respiratory infections are a common cause of hospital admissions and a common form of healthcare-associated infections. In both scenarios, early and appropriate antimicrobial therapy is required, which is often based on culture and susceptibility results, more so in the case of hospital-acquired pathogens having antimicrobial resistance (AMR) of various degrees. Additionally, ruling out or ruling in viral infections is equally essential. This calls for a syndromic approach toward diagnostics: searching for a wide range of pathogens contributing to the clinical findings. With the current inclination toward rapid diagnostics, molecular techniques

Address correspondence to Purva Mathur, purvamathur@yahoo.co.in.

The authors declare no conflict of interest.

with short specimen processing and turnaround times are being developed and utilized increasingly. Biofire Filmarray is an automated, single cartridge multiplex-PCR-based platform for testing species-specific targets and essential antimicrobial resistance genes from clinical samples. Biofire Filmarray Pneumonia plus Panel (BFP) identifies 15 frequently encountered Gram-negative and Gram-positive bacteria in semi-quantitative bacterial identification numbers (BIN), and atypical bacteria, i.e., *Legionella pneumophila*, *Mycoplasma pneumoniae*, and *Chlamydia pneumoniae* qualitatively. BFP also covers nine viruses causing pneumonia/lower respiratory tract infections, reported qualitatively. The AMR genes detected are KPC, NDM, Oxa-48-like, VIM, IMP and CTX-M (carbapenemase/ESBL), and mecA/mecC and MREJ for methicillin resistance.

The overall sensitivity and specificity stated by the manufacturer are 96.2% and 98.3%, respectively, for bronchoalveolar lavage (BAL)-like samples, and 96.3% and 97.2%, respectively, for sputum samples (1). In this study, we have evaluated BFP in a subset of indoor patients in wards and intensive care units (ICUs) at our tertiary care hospital by comparing it with standard microbiological culture and identification protocol (2).

## MATERIALS AND METHODS

### Study design

A one-and-a-half-year prospective study from January 2020 to June 2022, including patients admitted in wards/ICUs of a tertiary-level hospital in north India, aimed at comparing BFP with standard culture procedure.

### Criteria for accepting respiratory samples for BFP

Patients with deteriorating respiratory status, positive chest X-ray findings, and physician requests for BFP were included. Physicians usually requested BFP when they either suspected a non-bacterial etiology of pneumonia, bacterial cultures were non-contributive to diagnosis, or the patients were deteriorating clinically. Samples with ≥25 squamous epithelial cells per low power field (LPF) were rejected.

### Specimen processing and BFP

Sputum, endotracheal aspirate (ETA), and BAL samples were subjected to Gram's staining and inoculated onto 5% sheep blood agar, chocolate agar, and MacConkey agar semi-quantitatively and incubated at 37°C overnight. Growth of $\geq 10^4$ CFU/mL for BAL and $\geq 10^5$ CFU/mL for sputum and ETA was processed further (2). Identification and antimicrobial susceptibility testing (AST) of bacterial isolates were carried out using Vitek 2 (BioMérieux, Marcy-l'Etoile, France). Testing by Biofire Filmarray Pneumonia plus Panel was done per the manufacturer's instructions. The BFP reports the presence of an organism with "BIN," also called "binned values," ranging from $10^4$ to $10^7$ and the presence or absence of AMR genes. The BIN determined by BFP is a category corresponding to the approximate gene copies per milliliter. The standard microbiological culture was considered a gold standard. The pathogens and AMR genes detected in BFP are shown in Table 1.

### Polymerase chain reaction for AMR genes

The performance of BFP was evaluated against in-house polymerase chain reaction (PCR) detection of genes. ESBL (CTX-M) and carbapenemase genes NDM, OXA, and KPC were detected by PCR using the primers and thermocycling conditions from a previously published study by our group (3).

### Statistical analysis

GraphPad 7.0 and MS Excel (Microsoft Inc.) were used for statistical analysis. Categorical variables were expressed as numbers (%) and compared by the $\chi^2$ test or Fisher's exact

**TABLE 1** List of pathogens and AMR genes identified in BFP

| Pathogens | |
|---|---|
| Gram-negative (fermenters/nil-fermenters) | *Acinetobacter-calcoaceticus-baumannii* complex, *Enterobacter cloacae* complex, *Escherichia coli, Klebsiella aerogenes, K. oxytoca, K. pneumoniae, Proteus* spp., *Serratia marcescens, Pseudomonas aeruginosa* |
| Fastidious | *Haemophilus influenzae, Moraxella catarrhalis* |
| Gram positives | *Staphylococcus aureus, Streptococcus agalactiae, S. pneumoniae, S. pyogenes* |
| Atypical | *Chlamydia pneumoniae, Legionella pneumophila, Mycoplasma pneumoniae* |
| Viruses | Adenovirus, coronavirus, human metapneumovirus, human rhinovirus, influenza A, influenza B, middle east respiratory syndrome coronavirus (MERS CoV), parainfluenza, respiratory syncytial virus |
| AMR[a] genes | |
| CTX-M, IMP, KPC, NDM, OXA-48-like, VIM, mecA/C, and MREJ | |

[a]Antimicrobial resistance (AMR) genes.

test among multiple groups, age was presented as mean ± standard deviation (SD), and range. A *P* value <0.05 was considered statistically significant.

## RESULTS

### Patient details

Samples for BFP testing were received from patients admitted to the Departments of Medicine, Pulmonary Medicine, Cardiothoracic and Vascular Surgery, Gastroenterology, and Trauma Center. There were 94 (94/162, 58%) men and 68 (68/162, 42%) women in the age range of 15–82 years (mean = 48 years ± SD 18.3 years; median = 51 years). Samples consisted of endotracheal aspirate ($n$ = 62, 38.3%) and BAL ($n$ = 47, 29%) from mechanically ventilated patients and sputum ($n$ = 53, 32.7%) from the remaining.

### Overall detection of bacterial pathogens

One or more bacterial targets were detected in 128 samples, out of which 37 samples had either no growth or growth of upper respiratory flora and three samples had growth of bacteria outside the BFP panel (two *Stenotrophomonas maltophilia* and one *Burkholderia cepacia*). These three were removed from further analysis. Eighty-eight samples (88/159, 55.3%) had growth of one or two pathogens in significant quantity. The remaining 31 samples (31/159, 19.5%) had both negative BFP results and no growth in culture. This is detailed in Table 2.

Since we took culture as the gold standard, a positive BFP result with no growth/upper respiratory flora/growth outside the pneumonia panel was considered false positive. In contrast, a negative BFP result with growth in culture was considered false negative. Positive percent agreement (PPA) was calculated as [true positive/(true positive + false negative)] × 100%, and negative percent agreement (NPA) was calculated as [true negative/(true negative + false positive)] × 100%. The positive percent agreement was 100% (88/88), the negative percent agreement was 47.8% (34/71), and overall

**TABLE 2** BIOFIRE FILMARRAY pneumonia plus panel (BFP) versus culture

| | | Culture | | |
|---|---|---|---|---|
| | | Corresponding positive growth | No growth/URF[a] | Growth of organism not in panel[b] |
| BFP | ≥1 bacterial target detected | 88 | 37 | 3[b] |
| | No target detected | 0 | 34 | 0 |

[a]URF, upper respiratory flora.
[b]Samples with growth of "not in panel" organisms were excluded from the analysis.

agreement was 76.72% (122/159), and a kappa value of 0.504 [95% confidence interval (CI) 0.383–0.626], suggesting moderate agreement.

## Non-fastidious bacteria

The common pathogens detected in BFP were *Acinetobacter baumannii* complex/ *Acinetobacter baumannii-calcoaceticus complex* (*n* = 88), *Klebsiella pneumoniae* (*n* = 88), and *Pseudomonas aeruginosa* (*n* = 49).

### *Haemophilus influenzae*, *Streptococcus pyogenes*, and *Streptococcus pneumoniae*

*H. influenzae, S. pneumoniae,* and *S. pyogenes* were detected in 13 (8.1%), 4 (2.5%), and 2 (1.2%) samples, respectively.

## Agents of atypical pneumonia

*Mycoplasma pneumoniae* was detected in one sample, with another sample demonstrating *Chlamydia pneumoniae* to be present.

## Viruses

Viral pathogens were detected in 16 samples, of which human rhinovirus/enterovirus was the most common (10/16, 62.5%). Other viruses were coronavirus (*n* = 3), influenza A (*n* = 1), and parainfluenza virus (*n* = 1).

## Growth in culture with respect to BIN

Out of the 159 specimens processed, growth in culture was present in 88 (54.3%). Eleven (11/159, 6.9%) had growth of two Gram-negative bacterial pathogens, making a total of 99 isolates. These were *Acinetobacter baumannii* (*n* = 38), *Klebsiella pneumoniae* (*n* = 34), *Pseudomonas aeruginosa* (*n* = 16), *Escherichia coli* (*n* = 6), *Enterobacter cloacae* (*n* = 2), and *Staphylococcus aureus* (*n* = 3). Comparing with BFP results, we found that most isolates had a BIN of $10^7$ (87.8%, 87/99). A BIN $10^6$ was seen in 8.1% (8/99) of the isolates. The remaining isolates had BIN $10^5$ (*n* = 3) and $10^4$ (*n* = 1).

## Comparison between BIN in BFP and growth in bacterial culture

The overall agreement between culture and high BIN ($10^7$) in BFP was 63.7%, whereas 27.2% for the lower BIN ($10^6$, $10^5$, $10^4$). The $\chi^2$ statistic of growth or no growth between BIN $10^7$ vs lower BIN was 97.51, *P* value <0.00001. The culture result (growth or no growth) for each BIN is shown in Table 3.

We also noted that BFP detected >1 bacterial target in most samples (126/159, 79.25%) and single bacterial targets in 33 (33/159, 20.75%) samples. Concordant growth occurred more frequently in samples with a single target detected, as shown in Table 4.

Comparing growth in culture concerning different BIN, we found that the odds of growth in culture were significantly higher for BIN $10^7$ and $10^6$ when only a single bacterial target was detected in the sample.

## Detection by BFP vs growth in culture for the common pathogens

### *Acinetobacter baumannii*

*A. baumannii* was detected in BFP in 88 (88/159, 55.34%) samples. Growth in culture occurred in 38 (38/88, 43.18%) out of them, thus, having a PPA of 100%, NPA of 63.96%, overall agreement of 67.9%, and kappa of 0.404 (95% CI 0.295–0.514) indicating moderate agreement.

**TABLE 3** Comparison of BIN numbers and bacterial cultures

| BIN no. | Bacterial growth in culture | |
| --- | --- | --- |
| | Growth+ | Growth− |
| $10^7$ | 87 (55.7%) | 69 (44.3%) |
| $10^6$ | 8 (18.6%) | 35 (81.4%) |
| $10^5$ | 3 (4.7%) | 60 (95.3%) |
| $10^4$ | 1 (1.3%) | 74 (98.7%) |
| ND[a] | 0 | 34 (100%) |

[a]No bacterial target detected.

## Klebsiella pneumoniae

*K. pneumoniae* was detected in BFP in 88 (88/159, 55.34%) samples, and growth in culture occurred in 34 (34/88, 38.63%) out of them. Thus, having a PPA of 100%, NPA of 56.34% overall agreement of 66.03%, and kappa of 0.360 (95% CI 0.253–0.467) indicates fair agreement.

## Pseudomonas aeruginosa

*P. aeruginosa* was detected in BFP in 49 (49/159, 30.81%) samples, and growth in culture occurred in 16 (16/49, 32.65%) out of them. Thus, having a PPA of 100%, NPA of 76.92%, overall agreement of 79.24%, and kappa of 0.402 (95% CI 0.256–0.547) indicates moderate agreement.

## H. influenzae, S. pneumoniae, and S. pyogenes

These organisms were detected in 13, 2, and 4 samples, respectively; however, none of these grew on culture.

## Vitek 2 AST results compared to AMR genes detected by BFP

AST was performed by Vitek 2 for all the samples that had growth in culture. In the BFP, AMR genes most frequently detected were CTX-M (72/88, 81.8%), NDM (77/88, 87.5%), and OXA-48-like (65/88, 73.8%). All three together were present in 56 (63.6%) samples. Either of the carbapenemase genes was present in 81 (81/88, 92%) samples.

## Carbapenem resistance

For comparing the BFP results with phenotypic AST, carbapenem resistance was compared to the detection of any of the carbapenemase genes. Since the BFP shows only the presence or absence of AMR genes directly from the samples, in samples with two isolates grown on culture, concordance was considered if either of them conformed to the susceptibility predicted by the AMR genes detected in BFP. The concordance is shown in Table 5.

This gave a PPA of 94.4% (68/72), NPA of 7.1% (1/14), and overall agreement of 80.2% (69/86). All four false negatives by BFP were *Acinetobacter baumannii*. On the other hand,

**TABLE 4** Comparison of culture results with single vs multiple target detection by BFP

| BIN no. | With single target detected | | With multiple targets detected | | Significance |
| --- | --- | --- | --- | --- | --- |
| | Growth+ | Growth− | Growth+ | Growth− | P value |
| $10^7$ | 14 | 2 | 73 | 67 | 0.006[a] |
| $10^6$ | 4 | 0 | 4 | 35 | 0.006[b] |
| $10^5$ | 0 | 6 | 3 | 54 | 1[b] |
| $10^4$ | 0 | 6 | 1 | 68 | 1[b] |

[a]$\chi^2$ test.
[b]Fisher's exact test.

**TABLE 5** AMR genes in BFP vs AST result of Vitek 2

| | | Carbapenem resistance by Vitek 2 | |
|---|---|---|---|
| | | **Resistant** | **Susceptible** |
| OXA-48-like/NDM/VIM/IMP in BFP | + | 68 | 13 |
| | − | 4 | 1 |

in the false positives, BFP detected multiple bacterial targets, of which, only the one with a higher BIN grew on culture.

### Third generation cephalosporin resistance

This was compared in those samples ($n = 3$) which had only CTX-M gene detected in BFP. The presence of lone CTX-M was concordant with third generation cephalosporin resistance in all of these.

### Comparison of in-house PCR for AMR genes with BFP results

In-house PCR for CTX-M, NDM, KPC, OXA-48, VIM, and IMP was performed for 35 samples, which grew only one isolate on culture. Comparing with in-house PCR, we got four types of results: (i) all the genes detected by in-house PCR were also identified by BFP, categorized as "full concordance"; (ii) some of the genes detected by in-house PCR were unidentified by BFP, or, vice-versa, but there was some overlap, this was categorized as "partial concordance"; (iii) the AMR genes were only detected by BFP, categorized as "BFP+/PCR−"; and (iv) the AMR genes were only detected by in-house PCR, categorized as "BFP−/PCR+" (see Table S1 in Supplement 1 and Table S2 in Supplement 2). The results showing 60% (21/35) concordance are detailed in Table 6.

### Growth of pathogens outside the BFP panel

Additional pathogens picked up by culture were *Stenotrophomonas maltophilia* ($n = 1$) in a young woman having cystic fibrosis and *Burkholderia cepacia* ($n = 2$) in one patient with coronavirus disease 2019 and another with chronic obstructive pulmonary disease (COPD). These pathogens are not present in BFP.

## DISCUSSION

We observed a higher frequency of Gram-negative pathogens, both in BFP and standard culture procedure of respiratory specimens, which is seen in most Indian hospitals (4) (ICMR report 2021). *A. baumannii* was the most common organism detected, both as a single pathogen and as one among multiple pathogens. With the higher proportion of mechanically ventilated patients in our study, who were also suspected of having ventilator-associated pneumonia (VAP), we expected a similar outcome of predominantly Gram-negative, hospital-acquired pathogens with multi-drug resistance (5–7). Additionally, the microbiology of pneumonia in developing countries differs from that in developed countries, where the commonly responsible microbes are *S. pneumoniae*, *S. aureus*, and *H. influenzae* (8–11).

We found a PPA of 100% (88/88), NPA of 47.8% (34/71), and an overall agreement of 76.72% (122/159) for bacterial targets detected in BFP and growth in culture of the same organism.

A wide variation in the agreement of BFP with culture, a PPA in the range of 88%–100 % and NPA of 73.2%–98.1%, has been reported in the literature because of the different patient demography, community-acquired vs hospital-acquired infections, antibiotic policy, comparison methods, and discrepant analysis (6, 8, 12–16). In our study, we did not perform discrepant analysis by a third method for the false positives; for bacteria detected in BFP, which failed to grow on culture, culture positive was considered as true positive, resulting in low NPA. However, culture could be negative due to initiation of

**TABLE 6** AMR genes in BFP vs in-house PCR

| Category | N |
|---|---|
| Partial concordance[a] | 14 |
| Full concordance[b] | 7 |
| BFP+/PCR−[c] | 12 |
| BFP−/PCR+[d] | 2 |

[a]All the genes detected by in-house PCR were also identified by BFP.
[b]Some of the genes detected by in-house PCR were missed by BFP, or vice-versa.
[c]The AMR genes were only detected by BFP.
[d]The AMR genes were only detected by in-house PCR.

empirical antimicrobial therapy in the patients, inherent low sensitivity, lower number of organisms, or delayed transport of specimens.

Looking deeper into the BFP results, we observed that most samples had multiple bacterial pathogens in different BIN; only about a fifth had a single bacterial pathogen, making a difference in what grows on culture. A higher proportion of concordant growth was documented when a single bacterial target was detected, a finding noted by other researchers as well (11). Overall, in samples with multiple targets detected, bacteria detected in BIN $10^7$ had significantly higher chances of growth on culture; however, in the case of detection of single bacteria, the odds of growth were significantly high for BIN $10^7$ and $10^6$. Among the commonly isolated pathogens, the overall agreement was higher for *A. baumannii* and *P. aeruginosa*, a finding also shared by Yoo et al. (15). A few pathogens, *H. influenzae*, *S. pneumoniae*, and *S. pyogenes* failed to grow on culture despite being detected in 13, 2, and 4 samples, respectively, in BFP. These bacteria have notably shown poor concordance in other studies comparing BFP and culture. Most researchers have addressed this issue and zeroed in on administering effective antimicrobial therapy (6, 15, 17, 18). BFP, a PCR-based assay, is expected to have a higher detection ability than culture.

We found 60% concordance between AMR genes detected in BFP and in-house PCR, including only those samples with the growth of a single pathogen. The evaluation of AMR gene detection reported in several studies shows conflicting results; Buchan et al. have documented 20% concordance between the identification of carbapenemase/CTX-M by the BFP with phenotypic AST, PCR testing was not performed (14); whereas Lee et al. were able to confirm two out of three carbapenemase targets detected with phenotypic AST5. Yoo et al. found a higher concordance (80%) with phenotypic AST and PCR combined; however, PCR was performed only for resolving "false positives" (15).

Compared to phenotypic AST (Vitek 2), overall agreement with NDM/OXA-48/KPC/IMP/VIM detection in BFP of 80% was found; however, this needs further evaluation. One should also be careful when *A. baumannii* is detected but no AMR gene is detected; firstly, because this hospital-acquired pathogen is unlikely to be susceptible, and secondly, because the OXA-23-like carbapenemases are not in the panel. On the other hand, the presence of CTX-M resulted in third-generation cephalosporin resistance in all three isolates.

Rapid diagnostics and diagnostic stewardship are the backbone for timely treatment and antimicrobial stewardship programs. In our experience, BFP-based diagnosis has several advantages, for example, extremely rapid turnaround time, ~2 hours from sample receipt to conveyance of report to clinicians. This has been lifesaving in a few rapidly deteriorating patients and/or had viral pneumonia. The multiplex nature of detection prevents wastage of time since, in most hospitals, bacteriology, virology, mycology, and atypical pathogens are detected in separate laboratories, requiring samples to be sent to different labs.

The disadvantage of BFP is its very high cost, which precludes its use as a first-line diagnostics in resource-limited settings. Also, in the presence of multiple organisms detected in BFP, it is not easy to ascertain which organism harbors which AMR genes. However, this does not make much difference from the treatment perspective since the

presence of NDM or OXA genes directly from the samples alerts the clinicians and helps them give early and appropriate treatment.

Another matter in question is that BFP is a card-based test; we can perform only one test at a time. This limits the throughput of the test. Therefore, patient selection/prioritization is needed in a multi-specialty/multi-ICU setting.

Lastly, the pneumonia guidelines now recommend that antibiotics be initiated as early as possible, based on evidence from several studies (19–23). Here, the rapid molecular methods could be applied to both guide antibiotic treatment of HAP/VAP in ICUs and further the goals of antibiotic stewardship (24, 25). This should be followed up by culture results for detailed AST results and the growth of outside-panel bacteria.

## Limitations

Antimicrobial resistance may be conferred by mechanisms other than the genes detected by the BFP, thus restricting the comparison.

## Conclusion

Careful interpretation of BFP results is needed when multiple targets are detected; however, the detection of AMR genes should be used to guide therapy, and it must be followed up by culture to look for the phenotypic AST results. Interpretation is relatively straightforward in single-target detection; however, this is not a common occurrence.

## ACKNOWLEDGMENTS

We would like to express our gratitude to the members of our laboratory team, Ms. Seema and Ms. Nishu, for meticulously performing the Biofire assay and sample processing.

We sincerely thank Ms. Rezi for compiling and maintaining the data securely.

The manuscript does not contain information related to research involving human subjects.

## AUTHOR AFFILIATIONS

[1]Department of Microbiology, AIIMS, New Delhi, India
[2]Department of Laboratory Medicine, JPNATC, AIIMS, New Delhi, India
[3]Department of Orthopaedic, JPNATC, AIIMS, New Delhi, India

## AUTHOR ORCIDs

Smriti Srivastava  http://orcid.org/0000-0002-7252-8993
Purva Mathur  http://orcid.org/0000-0003-4429-3688

## ADDITIONAL FILES

The following material is available online.

Open Peer Review

PEER REVIEW HISTORY (review-history.pdf). An accounting of the reviewer comments and feedback.

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
