## [Reviewer comments · Microbiology Spectrum]

Microbiology Spectrum

Utility of A Multiplex Pathogen Detection System Directly from Respiratory Specimens for Treatment and Diagnostic Stewardship

Smriti Srivastava, Neha Sharad, Vandana Kiro, Aparna Ningombam, Sharad Srivastav, Kamran Farooque, and Purva Mathur

Corresponding Author(s): Purva Mathur, All India Institute of Medical Sciences

Review Timeline:

Submission Date:	October 23, 2023
Editorial Decision:	November 29, 2023
Revision Received:	January 15, 2024
Accepted:	March 29, 2024

Editor: Monika Kumaraswamy

Reviewer(s): Disclosure of reviewer identity is with reference to reviewer comments included in decision letter(s). The following individuals involved in review of your submission have agreed to reveal their identity: Thomas P Mawhinney (Reviewer #2)

Transaction Report:

DOI: <https://doi.org/10.1128/spectrum.03759-23>

Re: Spectrum03759-23 (Utility of A Multiplex Pathogen Detection System Directly from Respiratory Specimens for Treatment and Diagnostic Stewardship)

Dear Prof. Purva Mathur:

Thank you for submitting your work to Microbiology Spectrum. Your manuscript exploring the utility of BioFire Pneumonia Panel in diagnostic stewardship and the treatment of pneumonia at a tertiary hospital in India was highly informative. We would be pleased to consider a revised version for potential publication provided that you can satisfactorily address the substantive concerns raised in the reviews (appended below). Additionally, as the Editor I ask that you extensively revise the introduction describing the BIOFIRE® FILMARRAY® Pneumonia plus Panel. It should not have significant similarities to the wording present on the bioMérieux product website and modifications would help avoid any suggestion of plagiarism.

Below you will find my comments, instructions from the Spectrum editorial office, and the reviewer comments.

Revision Guidelines

Sincerely,
Monika Kumaraswamy, MD, D(ABMM)
Editor
Microbiology Spectrum

Reviewer #1 (Comments for the Author):

This manuscript by Dr. Srivastava, et al., entitled "Utility of A Multiplex Pathogen Detection System Directly from Respiratory Specimens for Treatment and Diagnostic Stewardship" is well-written and supportive of the need to be cautious with regards to interpreting the BIOFIRE® FILMARRAY® Pneumonia plus Panel when multiple pathogens are identified to be present.

- 1) Consider, if within limits of words permitted in the ABSTRACT section, writing out some abbreviations within the abstract, especially PPA and NPA (i.e., ... 100% positive percent of agreement (PPA) and 45.9% negative percent agreement (NPA)...). This is especially noted, as these are important enough to be defined within the manuscript text (lines 107-108), itself. This also applies to the first use of "AMR".
- 2) The presentation of the BIOFIRE® FILMARRAY® Pneumonia plus Panel, in the INTRODUCTION, is 'very similar' in wording to that found on the bioMérieux product website at BIOFIRE FILMARRAY Pneumonia plus Panel - clinical diagnostics products | bioMérieux Clinical Diagnostics (biomerieux-diagnostics.com). From this reviewer's perspective, this is acceptable given the appropriate reference (i.e, #1), but should include the completing words of the product line to include the following, to be accurate:
Line 42: "...and 9 viruses that cause pneumonia and other lower respiratory tract infections."
- 3) Line 105-106: Change "and" to "a" in "...,whereas and negative BFP result..." to "...,whereas a negative BFP result...."
- 4) Lines 119, 168, and 236: insert commas before the word "respectively".
- 5) Lines 121-122: Reads a bit awkwardly. Consider "Mycoplasma pneumoniae was detected in one sample with another sample demonstrating Chlamydia pneumoniae to be present."
- 6) Line 67: Consider modifying this sentence to read "The BFP reports the presence of an organism with "BIN numbers", also referred to as "binned values", that range from 104 to 107 and the presence or absence of AMR genes."

Reviewer #2 (Comments for the Author):

Mathur et al. explore the effectiveness of the BioFire Pneumonia plus Panel (BFP) in detecting respiratory pathogens and antimicrobial resistance markers in critically ill patients with acute respiratory distress. Conducted as a single-center study in a tertiary hospital in India, the research focuses on a setting with a high prevalence of multidrug-resistant Gram-negative infections. However, this specific clinical environment limits the generalizability of the findings. The authors found that the BFP displayed a high Positive Percent Agreement (PPA) with conventional culture methods, but had a lower Negative Percent Agreement (NPA), particularly effective in identifying fastidious organisms. The study also notes greater concordance in samples with higher Bacterial Identification Numbers (BIN) or when a single organism was detected. However, the presentation of data in the study is somewhat unclear and could benefit from further clarification.

Major Comments:

1. Table 2 is confusing particularly because of the No Growth/Upper respiratory flora/Outside panel bacteria. This reviewer does not think it is appropriate to classify off-panel bacteria such as *S. maltophilia* and *B. cepacia* along with the negative results. This will false scrow the PPA to 100% because the authors are only considering on-panel organisms for positive agreement. There could be a 3rd column for discrepant results (i.e. organisms detected by BFP and culture that are discrepant)
2. In table 4 the authors used the chi-square statistical test for BIN numbers of 107 and the Fisher exact test for lower number BINs. Should not the authors use one or the other?
3. Table 6 (AMR genes in BFP vs in-house PCR) is confusing to follow and should include a caption. The authors should list the AMR genes and in-house PCR results in the supplementary material.
4. The full data-set with each result should be made available as a supplementary material

Minor Comments:

1. The authors list *S. pneumoniae* and *S. pyogenes* as fastidious but these organisms are not typically considered fastidious.
2. There are spelling errors throughout.

Other Comments:

The authors conclusion that BFP results with multiple targets need to be interpreted with caution as there was less concordance for polymicrobial results as well results with lower BIN numbers. The representation of their data could be misleading, though. The 2x2 analysis of BFP vs culture results are particularly confusing. It is confusing to combine no growth, upper respiratory flora, and off-panel growth together. It also looks like they classified off-panel culture growth as negative, which is misleading and could represent infection in patients admitted with ventilatory associated pneumoniae. The writing is easy to follow but there are some spelling and grammar errors.

RESPONSE TO REVIEWERS

Reviewer #1:

1) Consider, if within limits of words permitted in the ABSTRACT section, writing out some abbreviations within the abstract, especially PPA and NPA (i.e., ... 100% positive percent of agreement (PPA) and 45.9% negative percent agreement (NPA)...). This is especially noted, as these are important enough to be defined within the manuscript text (lines 107-108), itself. This also applies to the first use of "AMR". –

Response: The mentioned expansions (PPA, NPA and AMR) have been added in the abstract.

2) The presentation of the BIOFIRE® FILMARRAY® Pneumonia plus Panel, in the INTRODUCTION, is 'very similar' in wording to that found on the bioMérieux product website at BIOFIRE FILMARRAY Pneumonia plus Panel - clinical diagnostics products | bioMérieux Clinical Diagnostics (biomerieux-diagnostics.com). From this reviewer's perspective, this is acceptable given the appropriate reference (i.e, #1), but should include the completing words of the product line to include the following, to be accurate:

Line 42: "...and 9 viruses that cause pneumonia and other lower respiratory tract infections."

Response: The content has been incorporated into the Introduction, which has also been modified as per editor's suggestions (lines 37-46, marked-up manuscript).

3) Line 105-106: Change "and" to "a" in "...,whereas and negative BFP result...." to "...,whereas a negative BFP result...."

Response: The change has been done

4) Lines 119, 168, and 236: insert commas before the word "respectively".

Response: The change has been done, line numbers are 129, 183 and 256 (marked-up manuscript)

5) Lines 121-122: Reads a bit awkwardly. Consider "Mycoplasma pneumoniae was detected in one sample with another sample demonstrating Chlamydia pneumoniae to be present."

Response: The lines have been modified as above (lines 132-133, marked-up manuscript)

6) Line 67: Consider modifying this sentence to read "The BFP reports the presence of an organism with "BIN numbers", also referred to as "binned values", that range from 104 to 107 and the presence or absence of AMR genes."

Response: The line has been modified as above (lines 77-79, marked-up manuscript).

Reviewer #2:

Major Comments:

1. Table 2 is confusing particularly because of the No Growth/Upper respiratory flora/Outside panel bacteria. This reviewer does not think it is appropriate to classify off-panel bacteria such as *S. maltophilia* and *B. cepacia* along with the negative results. This will false scrow the PPA to 100% because the authors are only considering on-panel organisms for positive agreement. There

could be a 3rd column for discrepant results (i.e. organisms detected by BFP and culture that are discrepant)

Response: The samples with growth of off-panel bacteria (n=3) have now been excluded from analysis, a separate column for these has been added to table 2.

2. In table 4 the authors used the chi-square statistical test for BIN numbers of 107 and the Fisher exact test for lower number BINs. Should not the authors use one or the other?

Response: In case of lower BIN numbers, one of the cells had “0” hence chi-square could not be done, and fisher exact test is accurate for smaller samples as compared to larger samples.

3. Table 6 (AMR genes in BFP vs in-house PCR) is confusing to follow and should include a caption. The authors should list the AMR genes and in-house PCR results in the supplementary material.

Response: Captions and legends have been added to Table 6, and the list of in-house PCR results (along with AMR genes detected in Biofire) has been added as Supplement-1.

4. The full data-set with each result should be made available as a supplementary material.

Response: The full data-set has been added as “Supplement-2”

Minor Comments:

1. The authors list *s. pneumoniae* and *S. pyogenes* as fastidious but these organisms are not typically considered fastidious.

Response: The term “fastidious” has been removed for *S. pneumoniae* and *S. pyogenes*.

2. There are spelling errors throughout.

Response: spell-check has been done.

Re: Spectrum03759-23R1 (Utility of A Multiplex Pathogen Detection System Directly from Respiratory Specimens for Treatment and Diagnostic Stewardship)

Dear Prof. Purva Mathur:

Your manuscript has been accepted, and I am forwarding it to the ASM production staff for publication. Your paper will first be checked to make sure all elements meet the technical requirements. ASM staff will contact you if anything needs to be revised before copyediting and production can begin. Otherwise, you will be notified when your proofs are ready to be viewed.

Please be sure to make the corrections recommended by the reviewer (see below).

Sincerely,
Monika Kumaraswamy, MD, D(ABMM)
Editor
Microbiology Spectrum

Reviewer #1 (Comments for the Author):

The authors have adequately addressed the comments from this reviewer.

I have the following minor comments:

1. In line 12 should the denominator be 87 instead of 88 (e.g. 38/87)?
2. Please add a legend or caption to table 2 to write out the abbreviation for URF (upper respiratory flora)